# Melatonin Induces Apoptosis and Modulates Cyclin Expression and MAPK Phosphorylation in Pancreatic Stellate Cells Subjected to Hypoxia

**DOI:** 10.3390/ijms22115555

**Published:** 2021-05-24

**Authors:** Matias Estaras, Manuel R. Gonzalez-Portillo, Miguel Fernandez-Bermejo, Jose M. Mateos, Daniel Vara, Gerardo Blanco-Fernandez, Diego Lopez-Guerra, Vicente Roncero, Gines M. Salido, Antonio González

**Affiliations:** 1Institute of Molecular Pathology Biomarkers, University of Extremadura, 10003 Caceres, Spain; meh@unex.es (M.E.); ramonglezpor@unex.es (M.R.G.-P.); gsalido@unex.es (G.M.S.); 2Department of Gastroenterology, San Pedro de Alcantara Hospital, 10003 Caceres, Spain; mfbermejo@unex.es (M.F.-B.); josemaria.mateos@salud-juntaex.es (J.M.M.); daniel.vara@salud-juntaex.es (D.V.); 3Hepatobiliary-Pancreatic Surgery and Liver Transplant Unit, University Hospital, 06080 Badajoz, Spain; gerardoblanco@unex.es (G.B.-F.); diego.lopezg@salud-juntaex.es (D.L.-G.); 4Unit of Histology and Pathological Anatomy, Veterinary Faculty, University of Extremadura, 10003 Caceres, Spain; roncero@unex.es

**Keywords:** apoptosis, fibrosis, hypoxia, mitogen-activated protein kinases, melatonin, pancreatic stellate cells

## Abstract

In certain diseases of the pancreas, pancreatic stellate cells form an important part of fibrosis and are critical for the development of cancer cells. A hypoxic condition develops within the tumor, to which pancreatic stellate cells adapt and are able to proliferate. The consequence is the growth of the tumor. Melatonin, the product of the pineal gland, is gaining attention as an agent with therapeutic potential against pancreatic cancers. Its actions on tumor cells lead, in general, to a reduction in cell viability and proliferation. However, its effects on pancreatic stellate cells subjected to hypoxia are less known. In this study, we evaluated the actions of pharmacological concentrations of melatonin (1 mM–1 µM) on pancreatic stellate cells subjected to hypoxia. The results show that melatonin induced a decrease in cell viability at the highest concentrations tested. Similarly, the incorporation of BrdU into DNA was diminished by melatonin. The expression of cyclins A and D also was decreased in the presence of melatonin. Upon treatment of cells with melatonin, increases in the expression of major markers of ER stress, namely BIP, phospho-eIF2α and ATF-4, were detected. Modulation of apoptosis was noticed as an increase in caspase-3 activation. In addition, changes in the phosphorylated state of p44/42, p38 and JNK MAPKs were detected in cells treated with melatonin. A slight decrease in the content of α-smooth muscle actin was detected in cells treated with melatonin. Finally, treatment of cells with melatonin decreased the expression of matrix metalloproteinases 2, 3, 9 and 13. Our observations suggest that melatonin, at pharmacological concentrations, diminishes the proliferation of pancreatic stellate cells subjected to hypoxia through modulation of cell cycle, apoptosis and the activation of crucial MAPKs. Cellular responses might involve certain ER stress regulator proteins. In view of the results, melatonin could be taken into consideration as a potential therapeutic agent for pancreatic fibrosis.

## 1. Introduction

Pancreatic cancer and pancreatitis are diseases that affect the pancreas and are characterized by the development of a fibrotic tissue that overgrows within the gland and progressively occupies most of the abnormal mass [1]. Activated pancreatic stellate cells (PSCs) are majorly involved in the process of fibrosis. Their increase in number, together with the release of signaling factors towards surrounding cells, creates an environment favorable for the growth of malignant cells. Under these circumstances, the growing tissue can evolve towards pancreatic cancer [2]. Moreover, fibrosis acts as a barrier against antitumor agents, thus emerging as an element of resistance in cancer therapy [3].

Due to the fast proliferation and accumulation of malignant cells within the tumor, the oxygen supply to the cells is compromised, and therefore, a condition of hypoxia develops [4]. Conversely to what would be expected from the low oxygen availability, the cells contained in the abnormal mass exhibit adaptation and survive. As a consequence, the tumor increases in size and can expand [5]. In relation to these observations, we have recently shown that PSCs proliferate under hypoxia. Similar to tumor cells, PSCs undergo certain changes that allow them to survive under a low oxygen supply [6]. Interestingly, the maneuvers developed by PSCs under low oxygen availability could contribute to fibrosis in the tumor and aid the growth of malignant cells [7].

Accumulation of metabolites that can enter signaling cascades and contribute to inflammation has been observed in hypoxia [8]. Additionally, tumor–stroma interactions are of key importance for inflammation within the gland and for the development of pancreatic ductal adenocarcinoma (PDAC), which exhibits a significant inflammatory response [9]. Therefore, it seems feasible that controlling the inflammatory response and the release of cytokines by cells forming the tumor mass might facilitate the treatments and restrain tumor growth. In this line, Porcelli et al. showed that activation of the TGF-β signaling in cancer-associated fibroblasts promoted tumor invasion [10]. The authors suggested that counteracting the high level of circulating proinflammatory/immunosuppressive cytokines might serve as a strategy for the treatment of PDAC. Thus, at present, it is widely accepted that a major challenge in the treatment of pancreatic cancer is the modulation of fibrosis development within the tumor.

Melatonin, the major product of the pineal gland, exhibits pleiotropic effects on cell physiology [11]. With time, research has given numerous pieces of evidence that signal its potential as a therapeutic agent. With respect to cancer disease, it has been shown that melatonin exerts antitumor action in brain cancer [12], breast cancer [13], colon cancer [14], lung cancer [15], liver cancer [16] and pancreatic cancer [17]. Interestingly, melatonin modulates the viability and proliferation of PSCs [18,19,20,21]; thus, its potential antifibrotic action should be explored.

Former research of our laboratory has revealed that the oxidative state of PSCs can be modulated by melatonin, which could influence their proliferative state. In addition, we have observed that PSCs exhibit adaptation to hypoxia and increase their proliferation (findings mentioned above). Because PSCs are majorly involved in the process of fibrosis that develops in pancreatic cancer and contribute to creating conditions that favor the growth of cancer cells, including hypoxic conditions, in the present work we have investigated the effects of melatonin on PSCs cultured under hypoxia. Bearing in mind the antitumoral actions of melatonin and the effects that we have observed on PSCs, we aimed to shed more light on the ways by which melatonin exerts its antiproliferative actions on PSCs. Clarification of the mechanisms involved in the actions of melatonin to modulate PSC physiology might help in the understanding of the ways by which pancreatic fibrosis could be resolved or diminished by melatonin. In this sense, putative actions of melatonin to decrease PSC proliferation could ease the treatment of pancreatic cancer. Thus, our major objective was to investigate whether melatonin might control the development of fibrotic tissue and might help in the therapy of cancer.

## 2. Results

### 2.1. Effect of Melatonin on Cell Viability and Proliferation

We have previously shown that PSCs proliferate under hypoxia [6]. Moreover, we have shown that melatonin decreases the viability of pancreatic cancer cells [22] and also of PSCs incubated under normoxia [19,20,21]. To study whether melatonin also decreases the viability of PSCs subjected to hypoxia, different preparations of PSCs were incubated under hypoxia for 48 h in the absence (nontreated cells) or in the presence of melatonin (1 mM, 100 µM, 10 µM or 1 µM). Separate batches of cells were exposed to thapsigargin (Tps; 1 µM), which served as control for cell death [23]. In the presence of melatonin, a statistically significant decrease in cell viability was observed at the higher concentrations used (1 mM and 100 µM). Treatment of cells with Tps, used as control for cell death, evoked a statistically significant decrease in cell viability (Figure 1A).

In the next step, we evaluated cell proliferation employing a kit based on 5-bromo-2-deoxyuridine (BrdU). BrdU incorporation into the DNA of dividing cells is an indicator of cell proliferation. In this set of experiments, cells were incubated for 48 h under hypoxia, in the absence (nontreated cells) or in the presence of melatonin (1 mM, 100 µM, 10 µM or 1 µM). Treatment of cells with melatonin induced a statistically significant decrease in BrdU content at the concentration of 1 mM. In the presence of the other concentrations of melatonin, slight decreases in BrdU content were observed, which were not statistically significant in comparison with that noted in nontreated cells (Figure 1B). Incubation of cells with Tps (1 µM) evoked a statistically significant decrease in BrdU content (Figure 1B).

Cyclins are a family of proteins with pivotal roles in the control of the cell cycle. In this part of the study, we were interested in analyzing whether melatonin exerts any effect on these proteins. Cyclin A is active in the early phases of division, and cyclin D regulates the transition from G1 to S phase [24,25].

The expression of cyclins was studied by Western blot. For this purpose, cells were incubated with melatonin (1 mM, 100 µM, 10 µM or 1 µM) for 4 h under hypoxia. Then, cell lysates were analyzed to determine the levels of cyclin A and cyclin D. The effects of melatonin on each cyclin are shown in Figure 1C–E. In general, melatonin induced a decrease in the expression of cyclins A and D. However, in the case of cyclin A, melatonin only decreased the level of protein at the concentrations of 1 mM and 100 µM. In the presence of Tps (1 µM), the detection of cyclins A and D was decreased (Figure 1C–E).

### 2.2. Effect of Melatonin on Endoplasmic Reticulum Stress

ER stress is a condition that develops in cancer and inflammation [26]. It has also been observed in viral infections and in metabolic, neurodegenerative and cardiovascular diseases [27]. BiP/GRP78 is an endoplasmic reticulum (ER) chaperone that plays a key role in the regulation of ER responses to stress. BiP exhibits antiapoptotic properties and has the ability to control the activation of transmembrane ER stress sensors such as IRE1, PERK and ATF6 [28].

In a first step, we incubated PSCs under hypoxia for 4 h in the absence of melatonin, and the levels of BiP, phosphorylated eIF2α and ATF-4 were studied by Western blotting. Under these conditions, we did not observe increases in the expression of the mentioned proteins. On the contrary, the detection of such proteins was decreased in comparison with the levels noted in cells incubated under normoxia (Appendix A).

Next, we investigated whether ER stress is involved in the responses of PSCs to melatonin treatment under hypoxia. For this purpose, cells were incubated under hypoxia and in the presence of melatonin (1 mM, 100 µM, 10 µM or 1 µM) for 4 h. Under these conditions, we detected an increase in the level of BiP (Figure 2A,B), in the phosphorylation of eukaryotic initiation factor 2 (eIF2α; Figure 2A,C) and in the level of ATF-4 (Figure 2A,D) in comparison with the levels noted in cells incubated in hypoxia but in the absence of melatonin. Treatment of cells with Tps (1 µM), an ER stress inducer [29] (Figure 4A,B), induced increases in the levels of the three proteins studied (Figure 2A–D).

### 2.3. Effect of Melatonin on Apoptosis

The decrease in cell viability that we have observed could be due to activation of apoptosis. Caspase-3 activation is a marker of apoptosis [22,30]. Therefore, we next evaluated the activation of caspase-3 in PSCs subjected to hypoxia and treated with melatonin for 24 h (1 mM, 100 µM, 10 µM or 1 µM). In the presence of melatonin, caspase-3 activation was observed (Figure 3). An increase in the level of activated caspase-3 was also noted in cells treated with the inducer of apoptosis Tps [31] (Figure 3).

### 2.4. Effect of Melatonin on MAPK Activation

The MAPK pathway plays a pivotal role in cellular signaling. This family of proteins transduces extracellular stimuli into phosphorylation events that control different cellular responses, which include inflammation, stress response, differentiation, survival and tumorigenesis [32].

Taking into account our results, we next evaluated the effect of melatonin treatment on MAPK activation in PSCs subjected to hypoxia. For this purpose, cells were incubated for 4 h under hypoxia and in the absence (nontreated cells) or in the presence of melatonin (1 mM, 100 µM, 10 µM or 1 µM).

Analysis of cell lysates revealed a statistically significant decrease in the phosphorylated state of JNK in cells treated with 1 mM melatonin. Conversely, an increase in phosphorylation was noted in samples from cells treated with the other concentrations of melatonin, although the differences were not statistically significant compared with the level detected in nontreated cells (Figure 4A,B). Melatonin treatment induced statistically significant increases in the phosphorylation of p38 and of p44/42 at all concentrations tested, in comparison with the values noted in nontreated cells (Figure 4A,C,D).

We next evaluated the involvement of p44/42 and p38 in the modulation of cell viability in cells treated with melatonin under hypoxia. Thus, PSCs were incubated for 5 min in the presence of U0126 (10 µM) and SB203580 (10 µM), which are specific inhibitors of p44/42 and p38, respectively. Thereafter, cells were incubated for an additional 48 h under hypoxia and in the presence of melatonin (1 mM–1 µM). Separate batches of cells were incubated in hypoxia and in the presence of the MAPK inhibitors. Cell viability was compared with that observed in nontreated cells (incubated in the absence of melatonin and without MAPK inhibitors). Inhibition of p44/42 and p38 significantly diminished viability of cells in comparison with nonstimulated cells. Moreover, treatment with the inhibitors decreased cell viability with respect to that noted in cells incubated in the presence of the respective concentration of melatonin (Figure 4E).

### 2.5. Effect of Melatonin on α-sma Expression

Alpha-smooth muscle actin (α-sma) is a specific marker for activated PSCs. Moreover, it has been signaled to exhibit an important role in fibrogenesis [33]. In this set of experiments, we studied the expression of α-sma in cells that had been incubated with melatonin (1 mM–1 µM) for 4 h under hypoxia. The results revealed a decrease in the content of α-sma in cells that had been treated with melatonin in comparison with cells incubated in its absence. A similar effect was noted in cells treated with 1 µM Tps (Figure 5).

### 2.6. Effect of Melatonin on Matrix Metalloproteinase Expression

A major contributor to the development of fibrosis is the secretion and accumulation of extracellular matrix components [34]. Matrix metalloproteinases (MMPs) are a family of proteins involved in processes such as angiogenesis, invasiveness and metastasis, which are considered important signs in cancer progression. The levels of members of this family of proteins are increased in pancreatitis and tumors [35]. Moreover, high levels of MMPs have been related to increased proliferation and migration of PSCs [36]. Because we had observed that melatonin induced changes in different biomarkers that are related to cell proliferation and that melatonin induced a decrease in cell viability, we decided to evaluate the effect of melatonin on the expression of MMP-2, MMP-3, MMP-9 and MMP-13.

PSCs were incubated for 4 h under hypoxia and in the presence of melatonin (1 mM, 100 µM, 10 µM or 1 µM). Separate batches of cells were incubated in the absence of melatonin but under hypoxia (nontreated cells). In general, treatment of cells with melatonin decreased the expression of MMPs, in comparison with the levels detected in nontreated cells (Figure 6). Tps (1 µM) also decreased the expression of MMPs (Figure 6).

## 3. Discussion

PSCs are resident cells of the pancreas. Normally inactive, under certain conditions, these cells undergo an activated state, and then a profibrogenic profile is set [37]. Particularly, PSCs have been pointed out as key contributors to the growth of pancreatic cancer, because of their contribution to stroma formation [38,39].

Remarkably, a special condition of hypoxia develops within abnormal tissues that exhibit a rapid development, as cancer tissues, due to the uncontrolled proliferation of cells contained in the mass [4]. In conjunction with cancer cells, PSCs will also be subjected to the low availability of O2 existing within the growing tumor. In order to survive, all types of cells forming part of the mass will exhibit adaptation to the low availability of O2 [40]. In this line, we have recently shown that PSCs adapt to hypoxia and exhibit increased proliferation capability with respect to PSCs grown in normoxia [6]. Under pathological conditions, such as inflammation and pancreatic cancer, PSCs contribute decisively to the development of the fibrotic reaction within the tissue. Therefore, maneuvers to reduce the amount of fibrotic tissue are considered a key tool in the treatment of these diseases [35,41,42]. In fact, it is now well accepted that the microenvironment contributes to the normalization of tumor cells. Hence, it might be feasible that modulation of stromal cells, rather than their removal, could be effective for cancer treatment [43].

Melatonin has been proposed to exhibit potential as a therapeutic agent in inflammation and cancer. With respect to cancer disease, it has been shown that melatonin exerts actions against a variety of tumors (for references, see introduction). As a part of the effects of melatonin, a decrease in the viability of cancer cells has been highlighted [44,45,46]. Its anticancer effects include the pancreas [22,47]. In addition, studies on PSCs have been conducted that show a decrease in cell viability in response to melatonin [18,20,21]. Moreover, PSCs exhibit adaptation to hypoxia and increases in their proliferation and migration ability [6]. However, the ways by which melatonin exerts its antiproliferative actions on PSCs are not completely understood, with special interest to hypoxia, and need further study.

Here we have shown that PSCs subjected to hypoxia exhibited a decrease in their viability after treatment with melatonin. These observations are in agreement with those reported formerly. In addition, our results have also shown that indoleamine decreases the expression of cyclins, which are pivotal proteins for the control of the cell cycle [24,25]. A clearer effect was exerted on cyclin D. A decrease in the expression of cyclin A was only noted with 1 mM and 100 µM melatonin. With regard to cyclin D, we have recently shown that PSCs subjected to hypoxia displayed an increase in its expression. This could be related to the increased proliferation that these cells exhibited [6]. Each of these cyclins plays a key role in the control of a certain step within the cell cycle. Interestingly, melatonin could regulate cell cycle via modulation of cyclin expression. In other words, those cells that survive upon melatonin treatment could exhibit a slowing down of the cell cycle, which might bring PSCs to a low proliferation rate. In relation to this, our results have shown that melatonin treatment diminished BrdU incorporation into DNA, which is an index of reduced cell proliferation. Therefore, our results suggest that melatonin could counteract the increased proliferation of PSCs that has been detected under hypoxia [6].

The unfolded protein response (UPR) is a condition that develops in cells subjected to stress, which promotes cell survival and adaptation to environmental conditions. The sensors PERK, IRE1 and ATF6 are pivotal to the UPR, together with their downstream transcription factors. When activated, the consequent UPR will trigger adaptation or apoptosis depending on the level of ER stress [48]. These proteins aim to restore homeostasis, but they can also induce cell death [49]. Melatonin has been shown to stimulate ER stress in different tumor cells, including the pancreas [50,51,52]. Our results have shown that PSCs subjected to hypoxia did not exhibit ER stress. However, upon treatment with melatonin, ER stress was observed. Our results are therefore in agreement with those previous observations.

Previous results of our laboratory showed that melatonin induced apoptosis in PSCs treated under normoxic conditions through caspase-3 activation [19]. This protein forms part of a regulated pathway that controls cell death, and its activation in cancer cells determines a decrease in cell viability [53]. Activation of caspase-3 by melatonin in pancreatic cancer cells has also been shown [22]. In the present work, we have shown that melatonin also induced the activation of caspase-3 under hypoxia. Moreover, the connection of ER stress with apoptosis has been signaled [54]. Thus, the induction of ER stress together with the activation of apoptosis that we have detected could represent two major pathways that might be recruited by melatonin in order to modulate the proliferation of PSCs. Interestingly, the cells that survive might enter the cell cycle in order to proliferate. However, melatonin might modulate the cell cycle to prevent the exacerbation of cell proliferation. This might be the reason why we observed a decrease in cell viability in the presence of melatonin, which was not as dramatic as that exerted by Tps; i.e., there is a remaining population of PSCs that survive and that could proliferate at a low rate, because the loss of cells was not complete following treatment with melatonin.

Another effect that we have observed is the activation of key components of the MAPK pathway in cells treated with melatonin. In a recent work, we showed that incubation of PSCs under hypoxia induced decreases in the phosphorylated state of p44/42 and p38, whereas an increase in the phosphorylation of JNK was noted [6]. In the present work, we have detected an increase in the phosphorylation of p38 protein. Activation of this MAPK is involved in cell death [55]. Moreover, it has been shown that melatonin-induced cell death involves phosphorylation of p38 [56]. Our results agree with these observations and suggest that the increase in p38 phosphorylation detected after treatment with melatonin could be related to the decrease in cell viability and proliferation that we have observed.

We also detected a decrease in the phosphorylation of JNK. This was only detected with the highest concentration of melatonin that we tested. Increases in its phosphorylation were observed at the other concentrations tested, but the differences were not statistically significant with respect to nontreated cells. Interestingly, 1 mM melatonin was the concentration that exerted the stronger effect on cell viability, cyclin expression and caspase-3 activation. It has been suggested that the JNK pathway plays a pivotal role in cell proliferation. JNK is phosphorylated in lung adenocarcinoma cells, hepatocellular carcinoma, colorectal cancer and pancreatic cancer, where it contributes to the progression of malignant cells and metastases [57,58,59,60,61]. Moreover, JNK phosphorylation is increased in PSCs subjected to hypoxia, and its inhibition is related to a decrease in cell viability [6].

In addition, we observed an increase in the phosphorylation of p44/42 in PSCs treated with melatonin. It is well known that this protein exerts a protective role [62]. From our point of view, the response that we have observed could be regarded as a counterpart by which, upon treatment with melatonin, PSC viability does not drop to an extent similar to that caused by Tps. In fact, increased phosphorylation of p44/42 by melatonin has been shown in HepG2 cells, a tumor cell line [56], where viability dropped upon melatonin treatment.

Our study suggests a parallel modulation of p44/42, p38 and JNK by melatonin, with probable competitive eventual consequences. In other words, handling of MAPK equilibrium could be an additional mechanism by which melatonin controls the proliferation of PSCs. Evidence in favor of this assumption is derived from the experiments in which cell viability dropped in cells treated with the inhibitors of p44/42 and p38.

It has been shown that activation of PSCs subjected to hypoxia occurs. A major marker of PSC activation is the expression of α-sma [63]. Interestingly, melatonin decreases the expression of this protein in myofibroblasts [64], endothelial cells [65], liver [66] or lung [67]. These observations support an antifibrogenic action of melatonin in these tissues. Our results have shown that melatonin induced a slight decrease in α-sma content. One possible explanation for the lack of a strong effect of melatonin on α-sma expression could be that melatonin does not completely reverse the activated state of PSCs subjected to hypoxia.

The development of fibrosis as a consequence of hypoxia is documented in pancreatic cancer [68]. MMPs comprise a multigene family of endopeptidases that are involved in remodeling of extracellular matrix during fibrosis. Moreover, MMPs are implicated in pathological processes, including cancer [69]. The activity of these proteins plays a pivotal role in tumor growth, invasion and metastasis [70,71]. Recent findings of our group showed that PSCs subjected to hypoxia exhibited an increase in the expression of MMP-2 [6]. Now we have shown that melatonin induced decreases in the expression of MMP-2, MMP-3, MMP-9 and MMP-13 in PSCs subjected to hypoxia. It has been suggested that, under hypoxia, PSCs release molecules that can act on neighboring cells to modulate their physiology. This might be related to the growth and proliferation of malignancy [72]. In addition, the migration capability of PSCs depends on the release of signaling factors by themselves. Accumulation of substances released by activated PSCs towards the extracellular medium can stimulate the activation of additional PSCs and other cells in the neighborhood. Because these proteins play a pivotal role in fibrosis and in cancer progression, our results show evidence in favor of a potential antifibrotic action of melatonin. A decrease in the secretion of components that could influence the remodeling of the extracellular matrix might be the basis of a putative action of melatonin to reduce the progression of abnormal tissue within the pancreas under hypoxia.

Taking into account the slight drop in cellular proliferation that was noted in the presence of melatonin under hypoxia, which might be related to the changes observed in the other biomarkers that we have studied, we could argue that melatonin might induce a putative lower activated state of PSCs. However, we cannot assure that cells reached complete quiescence. In a previous study by Estaras et al. [6], we showed that proliferation of PSCs was increased by 29% and that the expression of cyclin D was increased by 50% in cells subjected to hypoxia only, in comparison with the values detected in cells incubated in normoxia. Furthermore, the phosphorylation of JNK was increased in hypoxia by 20%, whereas the phosphorylation of p38 and of p44/42 dropped by 38% and 33%, respectively, with respect to the levels detected in cells incubated in normoxia. Additionally, in PSCs incubated under hypoxia, the expression of MMP-2, MMP-3, MMP-9 and MMP-13 was increased by 63%, 236%, 18% and 98.5%, respectively, in comparison with the values achieved in cells incubated in normoxia. All these previous observations are interesting, because our present results show effects of melatonin that are contrary to those observed in PSCs subjected to hypoxia only [6] and, hence, might support potential proquiescent effects of melatonin.

Altogether, our observations agree with the antiproliferative effects of melatonin that have been shown in pancreatic tumor cells [22] and in activated PSCs [19,20,21], the latter being cultured under normoxic conditions. Moreover, our results suggest that these effects also occur in hypoxia, a condition under which PSCs proliferate actively. The concentrations of melatonin that we have used fall within a range over that found in blood and, therefore, could be considered pharmacological and not physiological. However, melatonin is synthesized in various tissues, including the gastrointestinal system, where it could work in an autocrine or paracrine manner. This means that, in these tissues, melatonin could reach concentrations that are higher than those found in blood. Accordingly, the concentrations of melatonin found in blood cannot strictly define the local concentrations of melatonin that are considered physiological, because the latter have not been defined to date [73,74,75,76]. Concentrations of melatonin in the range of those employed in the present study have been used previously in healthy cells. In pancreatic acinar cells, melatonin stimulated the synthesis of antioxidant enzymes through Nrf2. This was considered a protective action of melatonin on healthy cells [77]. Additionally, melatonin protected pancreatic acinar cells against overstimulation with the secretagogue cholecystokinin, avoiding accumulation of Ca^2+^ in the cytosol and modulating amylase release [78,79]. Thus, pharmacological concentrations of melatonin might be useful in the therapy of pancreatic illnesses. Hypothetically, our results might have translational value to the clinic. However, in vivo studies need to be conducted to ascertain whether melatonin exerts the effects that have been observed in vitro.

The hypoxic response and the circadian clock exhibit reciprocal regulation. It has been suggested that hypoxia is gated by the circadian clock in vivo. In addition, hypoxia conversely regulates the clock genes by slowing the circadian cycle and smoothening the amplitude of oscillations. Hypoxia-inducible factor 1 is the key factor in charge of this regulation [80]. Moreover, metabolic adaptation to hypoxia elevates acid production within the tumor microenvironment. The acid produced during the cellular metabolic response to hypoxia suppresses the circadian clock through diminished translation of clock constituents. Suppression of the molecular clock’s oscillation affects the circadian transcriptome involving silencing of rapamycin complex 1 (mTORC1) signaling [81]. Another study revealed that O_2_ use and management of its byproducts (ROS) exhibit circadian variation due to changes in activity and metabolism during the day vs. the night. Thus, the metabolic products generated under hypoxia are among the most important physiological regulators of cellular differentiation/dedifferentiation. It is well known that blood flow is disrupted in tumors. As a consequence, transport of O_2_ or endocrine circadian regulators to the cells acquires key importance for circadian disruption and hypoxia in tumors. This is the reason why it has been suggested that circadian rhythms and hypoxia are involved in tumor growth and metastasis [82].

Finally, immune-based therapeutic strategies could be used to enhance PDAC cytotoxicity in order to restore immunity, which is disturbed in PDAC. In this line, the combination of melatonin with well-known immune-targeting agents hitting the microenvironment might reinforce the actions of treatments against fibrosis in PDAC [83]. In addition to immune modulation, other intracellular pathways could be used to modulate fibrosis. In this line, challenging the PI3K/mTOR pathway using inhibitors has been proposed to be a valuable tool for the treatment of cancer [84]. Modulation of the PI3K/mTOR pathway might enhance the therapeutic efficacy of melatonin by adding a significant antifibrosis effect. This could represent a line for future research.

## 4. Materials and Methods

### 4.1. Chemicals

Collagenase was obtained from Worthington Biochemical Corporation (Labclinics, Madrid, Spain). Cell Lytic for cell lysis and protein solubilization, crystal violet, protease inhibitor cocktail (Complete, EDTA-free), thapsigargin and Tween-20 were purchased from Sigma Chemicals Co. (Madrid, Spain). Fetal bovine serum, Hank’s balanced salts (HBSS), horse serum, medium 199 and SuperSignal West Femto were obtained from Fisher Scientific Inc. (Madrid, Spain). Polystyrene plates for cell culture were obtained from Thermo Fisher Sci. (Madrid, Spain). Penicillin/streptomycin was purchased from BioWhittaker (Lonza, Basel, Switzerland). Acrylamide, Bradford´s reagent, Tris/glycine/SDS buffer (10×) and Tris/glycine buffer (10×) were from Bio-Rad (Madrid, Spain). 5-Bromo-2-deoxyuridine (BrdU) cell proliferation assay kit was purchased from BioVision (Deltaclon S.L., Madrid, Spain). SB203580 and U0126 were obtained from Tocris (Biogen Científica, Madrid, Spain). Species-specific HRP-conjugated secondary antibodies were purchased from Thermo Fisher Sci. (Madrid, Spain). All other analytical-grade chemicals used were obtained from Sigma Chemicals Co. (Madrid, Spain).

### 4.2. Culture of Pancreatic Stellate Cells

PSCs were prepared and cultured using methods described previously. With the procedure employed, an enriched culture of activated PSCs with no contamination of other cell types was obtained [21]. The pancreas was obtained from Wistar rat pups (4–5 days after birth). Briefly, the pancreas was subjected to enzymatic digestion with a physiological buffer containing 130 mM NaCl, 4.7 mM KCl, 1.3 mM CaCl_2_, 1 mM MgCl_2_, 1.2 mM KH_2_PO_4_, 10 mM glucose, 10 mM HEPES, 0.01% trypsin inhibitor (soybean) and 0.2% bovine serum albumin (pH = 7.4 adjusted with NaOH) that was supplemented with 30 units/mL collagenase from Worthington. After centrifugation (30× *g* for 5 min at 4 °C) to remove the supernatant with the enzyme, culture medium was added to the pellet. Culture medium consisted of medium 199 supplemented with 4% horse serum, 10% FBS, a mixture of antibiotics (0.1 mg/mL streptomycin, 100 IU penicillin) and 1 mM NaHCO_3_. Next, mechanical dissociation of the cells was carried out by gently pipetting the cell suspension through tips of decreasing diameter. After centrifugation, cells were resuspended in culture medium. Finally, cells were seeded on different substrates depending on the studies to be carried out (round glass coverslips, 100 mm diameter Petri dishes, or multiwell polystyrene plates) and grown in a humidified incubator at 37 °C and 5% CO_2_. The experiments were carried out employing batches of cells obtained from different preparations. The number of passages of the cells was kept to a minimum (at most one passage was performed). The animals were supplied by the animal house of the University of Extremadura (Caceres, Spain). Handling of animals and the experimental protocols used were approved and performed according to the guidelines of the Ethical Committee for Animal Research of the University of Extremadura (identification code 44/2016; 14 July 2016) and the General Directorate of Agriculture and Livestock-Junta de Extremadura (identification code 20160810; 10 August 2016). The mentioned guidelines comply with EU bioethical law.

### 4.3. Induction of Hypoxia

Hypoxia was induced by incubation of cells in a low-O_2_ (1%) atmosphere [6]. An incubator chamber (Okolab; Izasa Scientific, Madrid, Spain) was employed. Temperature (37 °C), humidity (90%) and air atmosphere (content of 1% O_2_/5% CO_2_/94% N_2_) were electronically controlled.

### 4.4. Determination of Cell Viability and Proliferation

The crystal violet test was used to study the effect of treatments on cell viability [6]. Briefly, after treatment of cells with drugs, cells were washed with cold standard PBS and fixed with 4% paraformaldehyde (15 min at room temperature, 23–25 °C). Next, the cells were stained by incubation in the presence of 0.1% crystal violet (20 min at room temperature, 23–25 °C). This incubation was followed by washing with distilled water. Thereafter, water was removed, the wells were allowed to air dry and then 10% acetic acid was added to each well of the plate. Finally, the absorbance of each sample was measured at 590 nm employing a plate reader (CLARIOstar Plus, BMG Labtech., C-Viral, Madrid, Spain).

Cell proliferation was further analyzed by detection of BrdU incorporation into the DNA of growing cells as shown previously [19]. For this purpose, a commercially available kit (BrdU Cell Proliferation Assay Kit, from Biovision) was used. The protocol used was that suggested by the manufacturer. Absorbance of the samples was measured at 650 and 450 nm employing a plate reader (CLARIOstar Plus, BMG Labtech., C-Viral, Madrid, Spain).

Cells were subjected to drugs, and cellular viability or proliferation was compared with that of nontreated cells (subjected to hypoxia, but incubated in the absence of drugs). Data are shown as the mean change of absorbance expressed in percentage ± SEM (n) with respect to cells subjected to hypoxia in the absence of drugs (nontreated cells; n is the number of independent experiments).

Cell proliferation was further studied by detection of the expression of cyclin A and cyclin D and by detection of phosphorylation of MAPKs (p44/42, p38 and JNK). These determinations were carried out by Western blotting analysis, employing specific antibodies against respective proteins. Values are expressed as the mean ± SEM of normalized values expressed as percentage vs. the level of the respective protein found in cells subjected to hypoxia in the absence of drugs (nontreated cells; n is the number of independent experiments).

### 4.5. Determination of Apoptosis

Induction of apoptosis was detected by determination of capase-3 activation. Caspase-3 activation was determined employing previously described methods [19]. The cell-permeant substrate CellEvent Caspase-3/7 Green was used. This detection reagent consists of a four amino acid peptide (DEVD) conjugated to a nucleic acid binding dye. This cell-permeant substrate is intrinsically nonfluorescent because the DEVD peptide inhibits the ability of the dye to bind to DNA. After activation of caspase-3 in apoptotic cells, the DEVD peptide is cleaved, enabling the dye to bind to DNA and produce a bright, fluorogenic response with absorption/emission maxima of 502/530 nm. Samples were diluted in standard PBS to a final concentration 5 × 10^6^ cells /mL. The cells were stained with 1µL CellEvent Caspase-3/7 Green Detection Reagent (2 mM stock solution) and 1 µL of Hoechst 33342 (16.2 mM stock solution). After thorough mixing, the cell suspension was incubated at room temperature (23–25 °C) in the dark for 25 min; then, cells were loaded with 1 µL ethidium homodimer (1.167 mM in DMSO) and incubated for a further 5 min. Next, the samples were immediately run on the flow cytometer. The controls consisted of unstained and single-stained controls to properly set gates and compensations. Quantification of the fluorogenic response was performed by flow cytometry (Cytoflex flow cytometer; Beckman Coulter, Brea, CA, USA), with absorption/emission maxima of 502/530 nm. Unstained, single-stained, and fluorescence minus one (FMO) controls were used to determine compensations and positive and negative events, as well as to establish regions of interest. FlowJoV 10.4.1 Software (Ashland, OR, USA) was used for the analysis. Data are expressed in percentage of increase ± SEM (n) with respect to cells subjected to hypoxia in the absence of drugs (nontreated cells; n is the number of independent experiments).

### 4.6. Determination of Endoplasmic Reticulum Stress

Activation of endoplasmic reticulum (ER) stress leads to the expression of certain proteins that can be detected by Western blotting. Several markers of ER stress were detected: BiP/GRP78, phospho-eIF2α and ATF-4. Specific antibodies against the respective protein were used. Data are expressed in percentage ± SEM (*n*) with respect to cells subjected to hypoxia in the absence of drugs (nontreated cells; n is the number of independent experiments).

### 4.7. Western Blot Analysis

Western blotting analysis was employed for the determination of protein expression and/or phosphorylation, as described previously [22]. In brief, protein lysates (12 µg/lane) of each sample were separated by SDS-PAGE, using 10% polyacrylamide gels, and were transferred to nitrocellulose membranes. After blocking, the membranes were incubated overnight with the desired specific primary antibody. The list of primary antibodies that were used can be found in Table 1. After washing, the membranes were incubated for 1 h with the corresponding species-specific HRP-conjugated secondary antibody. Western blot was revealed and the bands were detected using Syngene GBOX Chemi-XX9 Gel Documentation System (C-Viral, Madrid, Spain). The software Image J (http://imagej.nih.gov/ij/; accessed on 4 December 2020) was used for quantification of the intensity of the bands [85]. Values are expressed as the mean ± SEM of normalized values expressed as percentage vs. nontreated cells.

### 4.8. Statistical Analysis

Normality of data was analyzed using Shapiro–Wilk test. Statistical analysis was performed by Mann–Whitney U test, and only *p* values < 0.05 were considered statistically significant. The software employed was GraphPad Prism (version 6.01). For individual comparisons and statistics between individual treatments, we employed the Student’s *t*-test, and only *p* values < 0.05 were considered statistically significant.

## 5. Conclusions

In summary, melatonin decreased the viability and proliferation of PSCs, which we have recently shown to be stimulated under hypoxia. Proliferating PSCs might contribute to the growth of malignant cells and, thus, could lead to the growth of abnormal tissue within a damaged pancreas. The effects that we have reported could be based on a proapoptotic action of melatonin. Moreover, a certain level of ER stress and major members of the MAPK family could be involved in the antiproliferative actions of melatonin. In addition, the cell cycle could be repressed, leading to a lower cellular proliferation. Finally, a decrease in the release of MMPs could contribute to the actions of melatonin. A summary of our findings can be seen in Figure 7. Because PSCs play a pivotal role in the fibrotic reaction that takes place in inflammation or cancer in the pancreas, where hypoxia exists, the conditions created by melatonin might slow down the proliferation of PSCs. This could control the fibrotic processes that can evolve under hypoxia and that could contribute to the survival and development of transformed epithelia within the pancreas. Our findings, therefore, support probable antiproliferative mechanisms by which melatonin could modulate fibrosis within the pancreas. Melatonin could thus serve as an aid in the strategies directed to control the development of fibrotic tissue within tumors and might help in the therapy of cancer.

## 6. Strengths and Limitations

Our results strengthen the hypothesis for a therapeutical use of melatonin in different diseases, including those that affect the pancreas, particularly inflammation and cancer. We provide evidence that sheds light on the effects of melatonin on the physiology of major cells that participate in the fibrotic processes that affect the gland. Moreover, our findings support former results obtained in other cell types of the pancreas, with which we have worked in the past: acinar cells, which were protected in the presence of melatonin, and tumor cells, which exhibited apoptosis in the presence of melatonin. The limitations of our study refer to the fact that we performed an in vitro study. In vivo studies need to be carried out in order to confirm the results obtained in vitro. Additionally, the involvement of other intracellular pathways and/or adaptations of metabolism in the cellular responses to melatonin needs to be further explored.

## Figures and Tables

**Figure 1 ijms-22-05555-f001:**
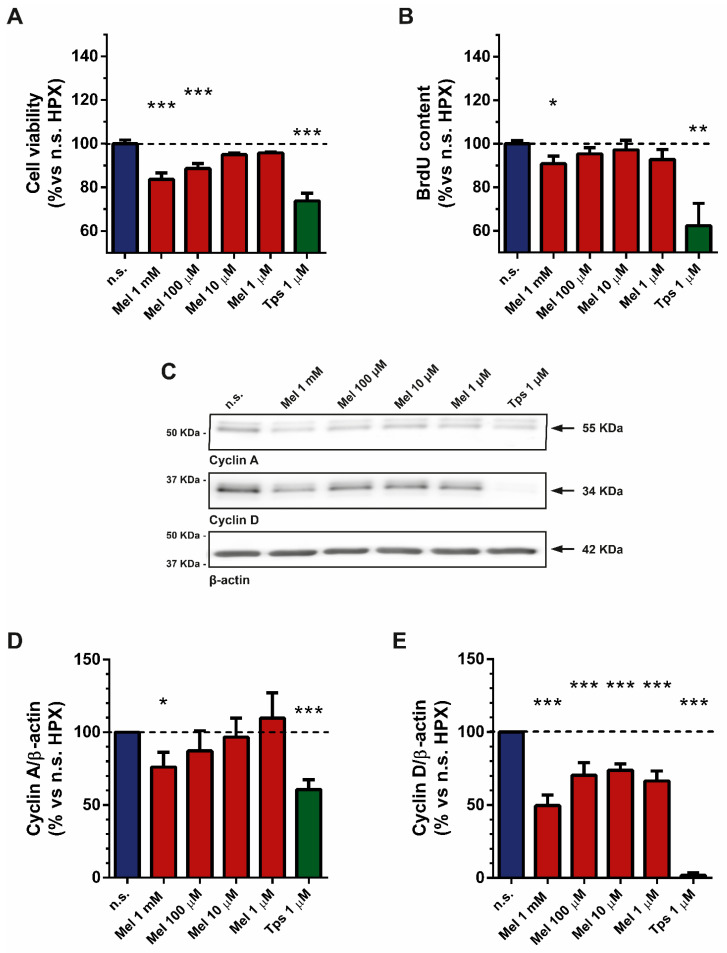
Effect of melatonin on cell viability and proliferation. (**A**) The bars show the effect of melatonin (1 mM, 100 µM, 10 µM or 1 µM) on the viability of PSCs incubated under hypoxia (in %: 86.77 ± 1.15; 94.16 ± 1.41; 97.38 ± 1.39; 98.39 ± 2.11; respectively for each concentration of melatonin vs. nontreated cells under hypoxia, which was considered 100%). Tps (1 µM) was used as control for cell death (in %: 63.67 ± 2.21). (**B**) The bars show the effect of melatonin on BrdU incorporation to DNA of dividing cells (in %: 90.88 ± 3.50; 95.36 ± 2.84; 97.19 ± 4.42; 92.81 ± 4.52; 62.38 ± 10.25; respectively for 1 mM, 100 µM, 10 µM or 1 µM melatonin and 1 µM Tps). (**C**) The figure shows representative blots of the expression of cyclin A and cyclin D, which were evaluated by Western blotting with specific antibodies. The band corresponding to each protein is marked by an arrow. The molecular weight of each specific protein is given on the right side of each blot. To ensure equal loading of proteins, the levels of β-actin were employed as controls under the tested conditions. (**D**,**E**) The bars show the quantification of protein levels for cyclin A (76.13 ± 10.01; 87.20 ± 13.91; 96.71 ± 13.01; 109.7 ± 17.56; 60.68 ± 6.79; respectively for 1 mM, 100 µM, 10 µM or 1 µM melatonin and 1 µM Tps) and cyclin D (49.69 ± 7.24; 70.57 ± 8.50; 73.90 ± 4.23; 66.61 ± 6.83; 1.79 ± 1.80; respectively for 1 mM, 100 µM, 10 µM or 1 µM melatonin and 1 µM Tps). Values show the mean ± SEM of normalized values expressed as % vs. nontreated cells (incubated under hypoxia and in the absence of melatonin or Tps), which was considered 100%. In the graphs, a horizontal dashed line represents the value achieved in nontreated cells. Data are representative of three to four independent experiments (n.s., nontreated cells; HPX, hypoxia; Mel, melatonin; Tps, thapsigargin; *, *p* < 0.05; **, *p* < 0.01; and ***, *p* < 0.001 vs. nontreated cells).

**Figure 2 ijms-22-05555-f002:**
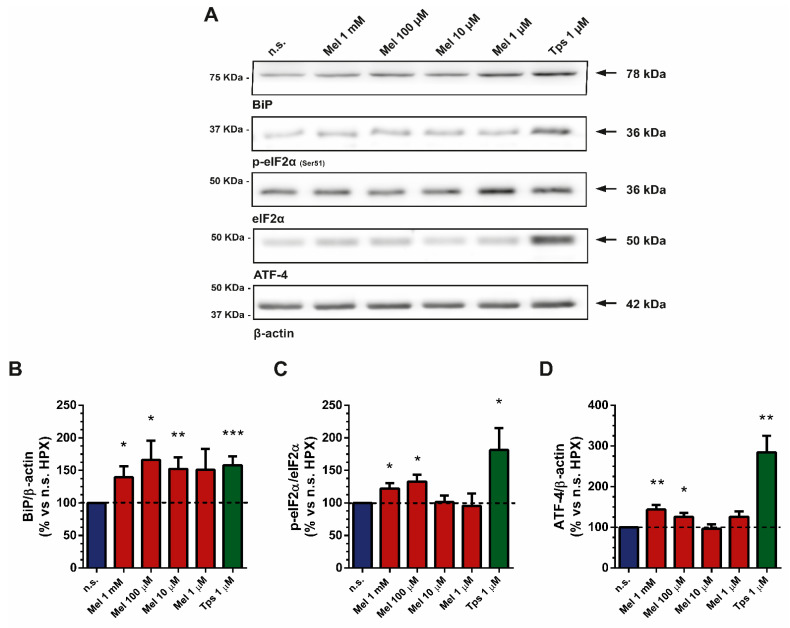
Effect of melatonin on ER stress markers in cells subjected to hypoxia. After treatment, cell lysates were processed for Western blotting analysis with specific antibodies. (**A**) Representative blots showing the effect of melatonin (1 mM, 100 µM, 10 µM or 1 µM) on the detection of the ER chaperone protein BiP/GRP78, the phosphorylation status of eIF2α and the level of ATF-4. The band corresponding to each protein is marked by an arrow. The molecular weight of each specific protein is given on the right side of each blot. To ensure equal loading of proteins, the levels of β-actin were employed as controls under the tested conditions for BIP and ATF-4, whereas the total expression level of eIF2α was used as control for p-eIF2α. (**B**–**D**) The bars show the quantification of protein levels for BIP (139.8 ± 16.64; 166.0 ± 29.82; 152.20 ± 17.94; 150.90 ± 32.34; 158.20 ± 13.37; respectively for 1 mM, 100 µM, 10 µM or 1 µM melatonin and 1 µM Tps), p-eIF2α (122.1 ± 7.90; 133.00 ± 10.57; 101.20 ± 10.30; 95.44 ± 19.29; 181.40 ± 33.74; respectively for 1 mM, 100 µM, 10 µM or 1 µM melatonin and 1 µM Tps) and ATF-4 (143.80 ± 10.93; 125.80 ± 9.83; 96.60 ± 11.03; 125.70 ± 13.38; 284.40 ± 40.88; respectively for 1 mM, 100 µM, 10 µM or 1 µM melatonin and 1 µM Tps). Values show the mean ± SEM of normalized values expressed as % vs. nontreated cells (incubated under hypoxia and in the absence of melatonin or Tps), which was considered 100%. In the graphs, a horizontal dashed line represents the value achieved in nontreated cells. Data are representative of four independent experiments (n.s., nontreated cells; HPX, hypoxia; Mel, melatonin; Tps, thapsigargin; *, *p* < 0.05; **, *p* < 0.01; and ***, *p* < 0.001 vs. nontreated cells).

**Figure 3 ijms-22-05555-f003:**
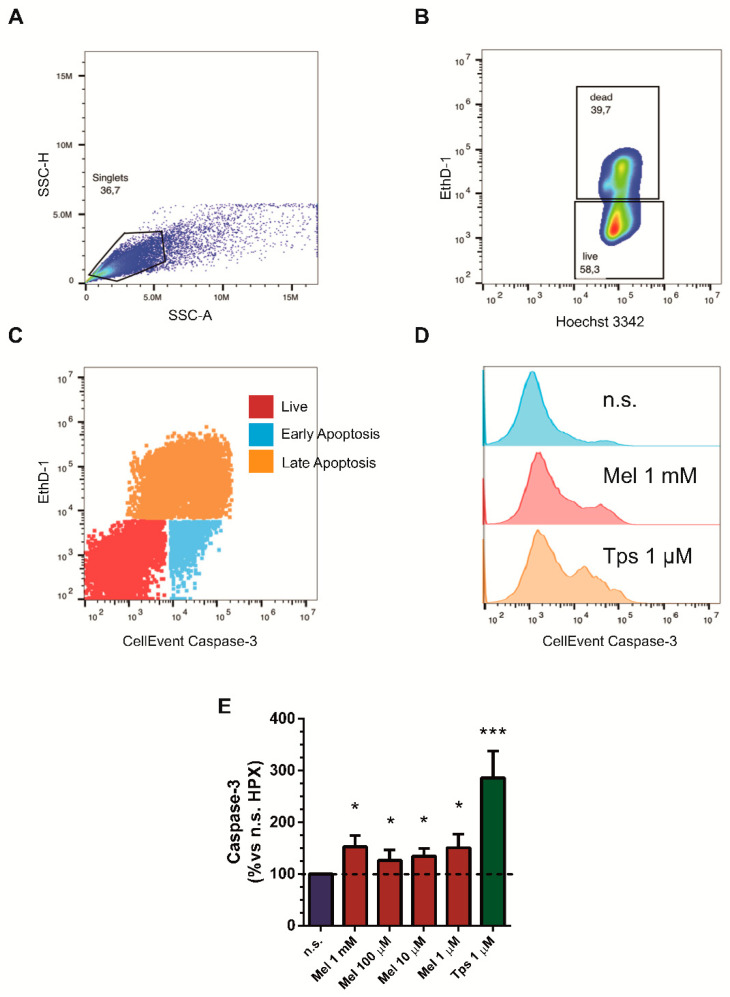
Study of caspase-3 activation in PSCs subjected to melatonin treatment under hypoxia. Cells were incubated for 24 h under hypoxia in the absence (nontreated) or in the presence of different concentrations of melatonin (1 mM, 100 µM, 10 µM or 1 µM) or with thapsigargin (Tps; 1 µM). (**A**) Doublets and clumps are identified and gated out to restrict the analysis to single cells. (**B**) 2D plot showing live and dead cells after staining with Eth-1 (dead) and Hoechst 3342 (live). (**C**) 2D dot plot showing apoptotic cells after staining with CellEvent and Eth-1. (**D**) The effect of the treatment is depicted, showing increased expression of caspase-3 from top to bottom. (**E**) The bars show the quantification of caspase-3 activation (152.80 ± 21.80; 126.40 ± 20.00; 134.40 ± 14.90; 150.70 ± 26.40; 285.90 ± 52.00; respectively for 1 mM, 100 µM, 10 µM or 1 µM melatonin and 1 µM Tps) in comparison with that detected in cells incubated in the absence of melatonin (nontreated cells), which was considered 100%. Tps (1 µM) was used as control. In the graph, a horizontal dashed line represents the level noted in nontreated cells (incubated under hypoxia and in the absence of melatonin or Tps), which was considered 100%. Results are representative of three different preparations (n.s., nontreated cells; HPX, hypoxia; Mel, melatonin; Tps, thapsigargin; *, *p* < 0.05; ***, *p* < 0.001 vs. nontreated cells).

**Figure 4 ijms-22-05555-f004:**
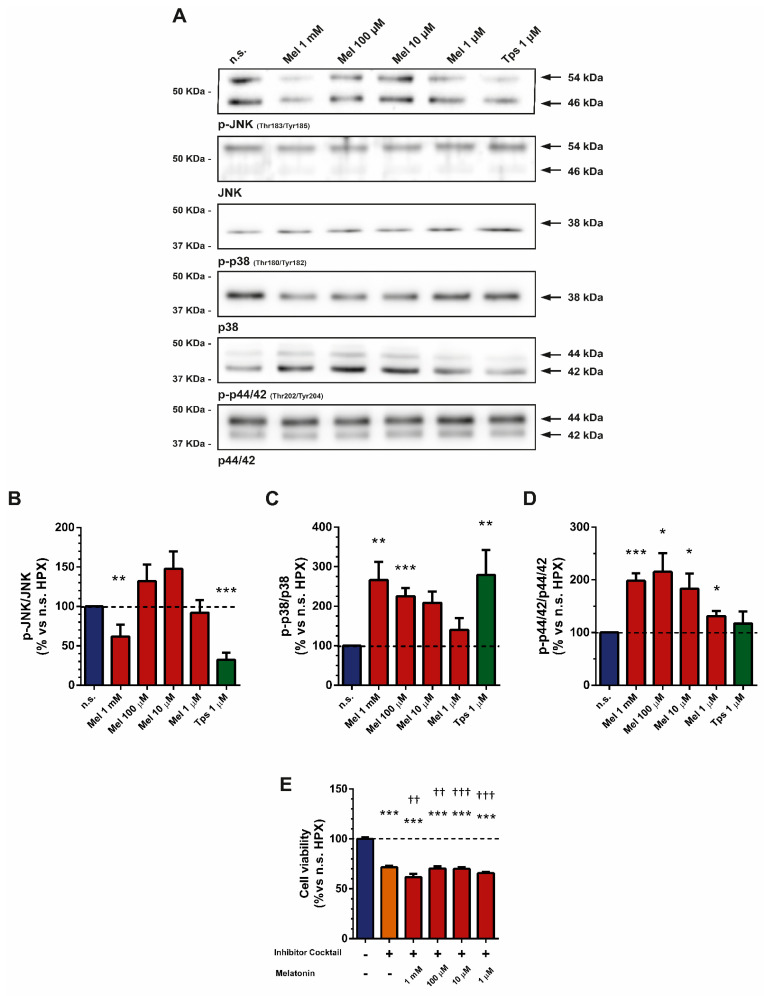
Analysis of JNK, p38 and p44/42 MAPK activation in response to melatonin under hypoxia. After treatment, cell lysates were processed for Western blotting analysis with specific antibodies against the phosphorylated form of each protein. (**A**) Representative blots showing the phosphorylation states of JNK, p38 and p44/42. The band corresponding to each protein is marked by an arrow. The molecular weight of each specific protein is given on the right side of each blot. (**B**–**D**) The bars show the quantification of protein phosphorylation for p-JNK (61.99 ± 15.09; 132.2 ± 21.21; 147.7 ± 22.09; 92.04 ± 16.27; 32.18 ± 9.016; respectively for 1 mM, 100 µM, 10 µM or 1 µM melatonin and 1 µM Tps), p-p38 (266.3 ± 45.96; 225.40 ± 20.73; 208.9 ± 28.02; 140.4 ± 29.78; 279.30 ± 62.77; respectively for 1 mM, 100 µM, 10 µM or 1 µM melatonin and 1 µM Tps) and p-p44/42 (198.70 ± 14.43; 215.30 ± 35.56; 183.1 ± 28.91; 131.00 ± 9.67; 117.1 ± 23.03; respectively for 1 mM, 100 µM, 10 µM or 1 µM melatonin and 1 µM Tps). A horizontal dashed line represents the value observed in nontreated cells (incubated under hypoxia and in the absence of melatonin or Tps), which was considered 100%. Values show the mean ± SEM of normalized values expressed as % of phosphorylation vs. nontreated cells. Data are representative of four to five independent experiments (n.s., nontreated cells; HPX, hypoxia; Mel, melatonin; *, *p* < 0.05; **, *p* < 0.01; and ***, *p* < 0.001 vs. nontreated cells). (**E**) The bars show the viability of cells incubated for 48 h under hypoxia and in the presence of different concentrations of melatonin (1 mM, 100 µM, 10 µM or 1 µM) in combination with a cocktail of specific inhibitors of p44/42 (U0126; 10 µM) and p38 (SB203580; 10 µM) (in %: 71.57 ± 1.52; 61.73 ± 3.36; 70.50 ± 2.07; 70.09 ± 1.59; 65.82 ± 1.14; respectively for inhibitor cocktail alone and inhibitor cocktail plus 1 mM, 100 µM, 10 µM or 1 µM melatonin vs. nontreated cells, which was considered 100%). The inhibitors were added to the cells 5 min prior to addition of the respective concentration of melatonin. Viability was compared with that of cells incubated in the absence of melatonin (nontreated cells; *) and with the respective concentration of melatonin (†). Tps (1 µM) was used as control for cell death. In the graph, a horizontal dashed line represents the mean value of viability of nontreated cells (incubated under hypoxia and in the absence of drugs), which was considered 100%. Data are representative of three independent experiments (n.s., nontreated cells; Inhib. cocktail, cocktail of p44/42 and p38 inhibitors; Mel, melatonin; Tps, thapsigargin; ***, *p* < 0.001 vs. nonstimulated cells; ††, *p* < 0.01; †††, *p* < 0.001 vs. the respective concentration of melatonin).

**Figure 5 ijms-22-05555-f005:**
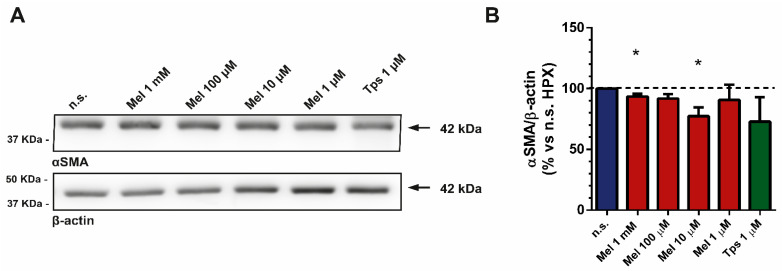
Analysis of α-smooth muscle actin expression in response to melatonin under hypoxia. After treatment, cell lysates were processed for Western blotting analysis with a specific antibody. (**A**) Representative blot showing the expression of α-sma. To ensure equal loading of proteins, the levels of β-actin were employed as controls under the tested conditions. The band corresponding to each protein is marked by an arrow. The molecular weight of each specific protein is given on the right side of each blot. (**B**) The bars show the quantification of α-sma expression (93.42 ± 2.30; 91.86 ± 3.52; 77.37 ± 7.16; 90.71 ± 12.49; 72.87 ± 20.01; respectively for 1 mM, 100 µM, 10 µM or 1 µM melatonin and 1 µM Tps). A horizontal dashed line represents the value observed in nontreated cells (incubated under hypoxia and in the absence of drugs), which was considered 100%. Values show the mean ± SEM of normalized values expressed as % vs. nontreated cells. Data are representative of three separate experiments (n.s., nontreated cells; HPX, hypoxia; Mel, melatonin; Tps, thapsigargin; *, *p* < 0.05 vs. nontreated cells).

**Figure 6 ijms-22-05555-f006:**
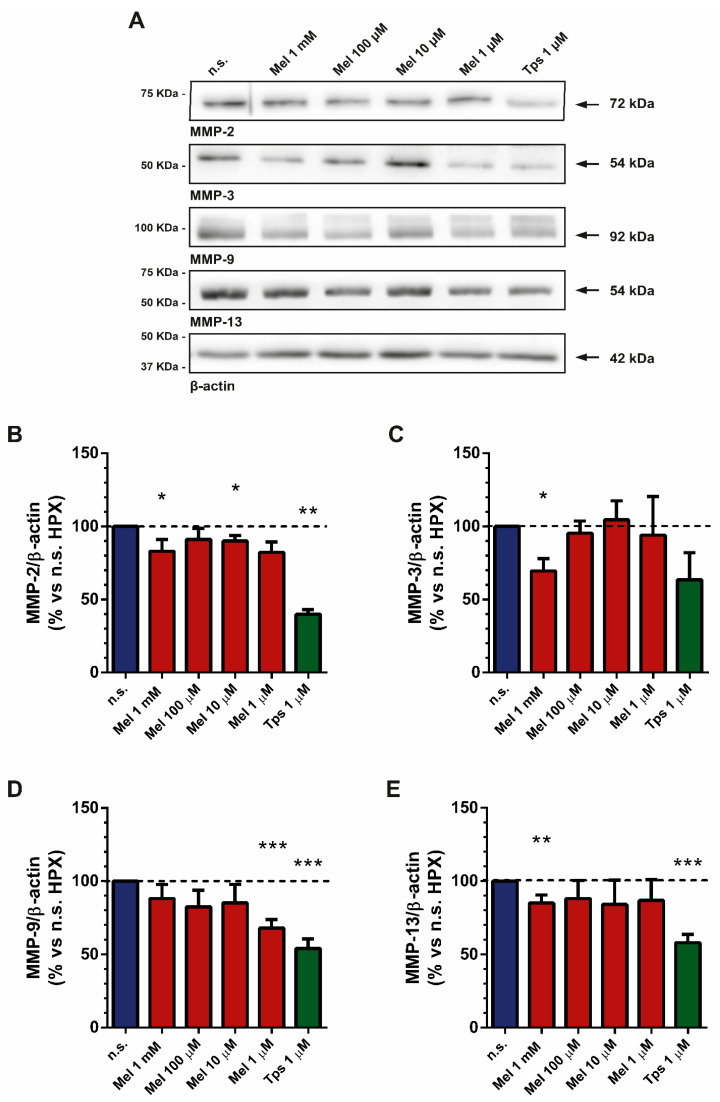
Analysis of MMP expression in response to melatonin under hypoxia. After treatment, cell lysates were processed for Western blotting analysis with specific antibodies. (**A**) Representative blots showing the expression of MMP-2, MMP-3, MMP-9 and MMP-13. To ensure equal loading of proteins, the levels of β-actin were employed as controls. The band corresponding to each protein is marked by an arrow. The molecular weight of each specific protein is given on the right side of each blot. (**B**) The bars show the quantification of protein expression for MMP-2 (83.12 ± 7.94; 91.20 ± 7.53; 90.11 ± 3.64; 82.29 ± 7.08; 39.98 ± 3.12; respectively for 1 mM, 100 µM, 10 µM or 1 µM melatonin and 1 µM Tps), MMP-3 (69.55 ± 8.51; 95.46 ± 8.22; 104.60 ± 12.99; 94.12 ± 26.45; 63.51 ± 18.61; respectively for 1 mM, 100 µM, 10 µM or 1 µM melatonin and 1 µM Tps), MMP-9 (88.03 ± 9.71; 82.43 ± 11.33; 85.19 ± 12.62; 67.94 ± 5.69; 54.12 ± 6.56; respectively for 1 mM, 100 µM, 10 µM or 1 µM melatonin and 1 µM Tps) and MMP-13 (85.08 ± 5.43; 88.03 ± 12.44; 84.03 ± 16.66; 86.81 ± 14.15; 58.09 ± 5.64; respectively for 1 mM, 100 µM, 10 µM or 1 µM melatonin and 1 µM Tps). A horizontal dashed line represents the value observed in nontreated cells (incubated under hypoxia and in the absence of drugs), which was considered 100%. Values show the mean ± SEM of normalized values expressed as % vs. nontreated cells. Data are representative of three to four separate experiments (n.s., nontreated cells; HPX, hypoxia; Mel, melatonin; Tps, thapsigargin; *, *p* < 0.05; **, *p* < 0.01; ***, *p* < 0.001 vs. nontreated cells).

**Figure 7 ijms-22-05555-f007:**
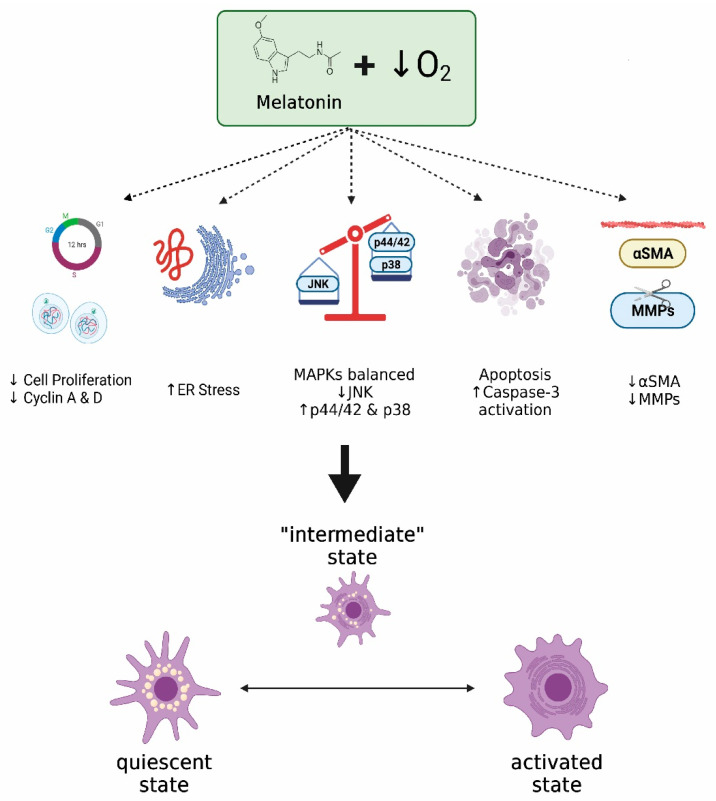
Summary of the effects of melatonin on PSCs subjected to hypoxia. PSCs exhibit increased proliferation under hypoxia. Thus, proliferating PSCs might contribute to the growth of abnormal tissue within a damaged pancreas, allowing the growth of malignant cells included in the mass. Melatonin may exert a certain level of ER stress and a proapoptotic action that could lead to a certain degree of cell death. In addition, melatonin could influence the cell cycle, thereby slowing down the proliferation of surviving cells. Major members of the MAPK family could be involved. Additionally, a decrease in the release of MMPs and a decrease in the expression of α-sma were also observed in cells treated with melatonin. Melatonin might therefore control the viability and proliferation of PSCs under hypoxia, leading the cells to a hypothetic “intermediate state”. Hence, melatonin might decrease the fibrotic reaction that could lead to impairment of the pancreatic function and that could contribute to survival and development of transformed epithelia within the pancreas (α-sma, α smooth muscle actin; ER, endoplasmic reticulum; JNK, c-Jun N-terminal kinase; MMP, matrix metalloproteinase). Figure created with BioRender software (BioRender.com).

**Table 1 ijms-22-05555-t001:** Primary antibodies used in the study.

Antibody	Dilution	Supplier
ATF-4	1:2000	Abcam
β-Actin HRP-conjugated	1:50000	Thermo Fisher
BiP	1:2000	Cell Signaling
Cyclin A	1:2000	Thermo Fisher
Cyclin D	1:10000	Abcam
MMP-2	1:2000	Abcam
MMP-3	1:2000	Abcam
MMP-9	1:2000	Abcam
MMP-13	1:2000	Abcam
p-eIF2α	1:2000	Abcam
eIF2α	1:1000	Cell Signaling
p-JNK	1:1000	Cell Signaling
JNK	1:1000	Cell Signaling
p-p38	1:1000	Cell Signaling
p38	1:1000	Cell Signaling
p-p44/42	1:2000	Cell Signaling
p44/42	1:2000	Cell Signaling
α-sma	1:1000	Thermo Fisher

The primary antibodies listed were specific for each protein. The detection of the desired protein was carried out by Western blotting analysis, as described in Section 4. Thermo Fisher (Madrid, Spain); Abcam plc (Cambridge, UK); Cell Signaling (C-Viral, Madrid, Spain).

## Data Availability

Data are available from the corresponding author upon reasonable request.

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
