# Peer review of "Melatonin Induces Apoptosis and Modulates Cyclin Expression and MAPK Phosphorylation in Pancreatic Stellate Cells Subjected to Hypoxia"

_ijms, 2021, doi:10.3390/ijms22115555_

Round 1

Reviewer 1 Report

Matias Estaras uncovered cellular responses to involve certain ER stress regulator proteins. In view of the results, the author pinpoint melatonin to be taken into consideration as potential therapeutic agent for pancreatic fibrosis. The manuscript covers an interesting topic well, nonetheless, there are few sections that deserve to be restructured, in order to achieve the level and comprehensive overview that the journal like IJMS would aim to.

Major points to consider in subsequent versions:

Introduction: In the introduction, the authors briefly summarize about the role of fibrosis in pancreatic homeostasis. This section can be slightly expanded, comprising novel findings of this topic and the insights about TGF-beta, leading to tumor stroma interactions are of key importance for pancreas inflammation within the pancreatic ductal adenocarcinoma (PDAC) and the physiological conditions (please refer to PMID: 30866547).

Results.

  1. for all Western blot figures, densitometry readings/intensity ratio of each band should be included; the whole Western blot showing all bands and molecular weight markers should be included in the Supplementary Materials. 
  2. Did the author employ unstained, isotype controls in the FACS analyses?

General comments: The authors' statement “As a part of the effects of melatonin, a decrease in the viability of cancer cells has been highlighted.” In ref 43 Wank highlight that Hypoxic Tumor-Derived Exosomal miR-301a Mediates M2 Macrophage Polarization via PTEN/PI3Kgamma to Promote Pancreatic Cancer Metastasis” is of interest, nonetheless, I personally miss some important translational aspect potentially related to this aspect, pointing towards a potential Achilles’ heel of PDAC that might be exploited therapeutically in the future. Indeed, while inhibiting hypoxic affected microenvironment, and PI3K/AKT pathway, combination with well-known immune-targeting agents hitting the microenvironment (refer to PMID: 31277479) andPI3K/ mTOR inhibitors (please refer to PMID: 29755672) might enhance the therapeutic efficacy by adding a significant anti-fibrosis effect to silencing of melatonin.

The figures would require some beautification and should be as simplified as possible in order to make them as much readable as possible.

Please correct typos (i.i. apoptosis instead of apopstosis at 4.5)

Author Response

Manuscript: ID ijms-1203381.

Title: Melatonin Induces Apoptosis and Modulates Cyclins Expression and MAPKs Phosphorylation in Pancreatic Stellate Cells Subjected to Hypoxia.

Authors: Matias Estaras et al.

Reply to the reviewer #1 comments

General comment: Matias Estaras uncovered cellular responses to involve certain ER stress regulator proteins. In view of the results, the author pinpoint melatonin to be taken into consideration as potential therapeutic agent for pancreatic fibrosis. The manuscript covers an interesting topic well, nonetheless, there are few sections that deserve to be restructured, in order to achieve the level and comprehensive overview that the journal like IJMS would aim to.

Reply: Thank you very much for reviewing our manuscript, for your suggestions and for drawing our attention to the different points raised.

Comment 1: In the introduction, the authors briefly summarize about the role of fibrosis in pancreatic homeostasis. This section can be slightly expanded, comprising novel findings of this topic and the insights about TGF-beta, leading to tumor stroma interactions are of key importance for pancreas inflammation within the pancreatic ductal adenocarcinoma (PDAC) and the physiological conditions (please refer to PMID: 30866547).

Reply: Thank you very much for this comment. We have included in the introduction information regarding the relationship of hypoxia and stroma with inflammation. New studies have been cited, including that suggested by the reviewer (lines 58-69). The new references have been included in the bibliography list.

Comment 2: For all Western blot figures, densitometry readings/intensity ratio of each band should be included; the whole Western blot showing all bands and molecular weight markers should be included in the Supplementary Materials.

Reply: Thank you very much for this observation. We have included in the legend to the figures the intensity ratio of each band shown in the Western blots. The original, uncropped and unadjusted images for blots were uploaded as Supporting Information files at the time of initial submission to the Journal, as required for the submission process. This supporting information has been now incorporated as Supplementary material in the revised version of the manuscript.

Comment 3: Did the author employ unstained, isotype controls in the FACS analyses?

Reply: The determination of Caspase-3 activation was carried out using CellEvent™ Caspase-3/7 Green Detection Reagent and was measured by flow cytometry. This reagent is based on a specific binding DNA dye linked to a short peptide (DEVD). This peptide avoids that the dye could bind to DNA. Under these circumstances, the reagent is intrinsically non-fluorescent. When apoptosis is activated, caspase-3 cleaves DEVD, enabling the dye to bind to DNA and then produces a bright, fluorogenic response. We did not use any antibody for the determination of caspase-3 activation. By this reason, we did not use any isotype control. Nevertheless, we employed a control of cells that were not incubated with the Cell-Event Reagent, to exclude cell autofluorescence in the determinations.

Comment 4: I personally miss some important translational aspect potentially related to this aspect, pointing towards a potential Achilles’ heel of PDAC that might be exploited therapeutically in the future. Indeed, while inhibiting hypoxic affected microenvironment, and PI3K/AKT pathway, combination with well-known immune-targeting agents hitting the microenvironment (refer to PMID: 31277479) andPI3K/ mTOR inhibitors (please refer to PMID: 29755672) might enhance the therapeutic efficacy by adding a significant anti-fibrosis effect to silencing of melatonin.

Reply: Thank you very much for this observation. We have found a mistake in Ref. 43 “Wang, X.; Luo, G.; Zhang, K.; Cao, J.; Huang, C.; Jiang, T.; Liu, B.; Su, L.; Qiu, Z. Hypoxic Tumor-Derived Exosomal miR-301a Mediates M2 Macrophage Polarization via PTEN/PI3Kgamma to Promote Pancreatic Cancer Metastasis. Cancer Res 2018, 78, 4586-4598, DOI 10.1158/0008-5472.CAN-17-3841”. The cyted work should be “Wang TH, Hsueh C, Chen CC, Li WS, Yeh CT, Lian JH, Chang JL, Chen CY. Melatonin Inhibits the Progression of Hepatocellular Carcinoma through MicroRNA Let7i-3p Mediated RAF1 Reduction. Int J Mol Sci. 2018 Sep 10;19(9). pii: E2687. doi: 10.3390/ijms19092687”. We have corrected the mistake, deleted the former reference and added the correct one (see ref. 46). We apologize for this mistake.

In addition, we have included in the discussion section comment to immune-targeting agents hitting the microenvironment and to the use of PI3K/ mTOR inhibitors which could exert significant anti-fibrosis effect, as the reviewer has suggested. The new text can be found in lines 480-488. The proposed bibliography has been cited and has been included in the bibliography list.

Comment 5: The figures would require some beautification and should be as simplified as possible in order to make them as much readable as possible.

Reply: Following the reviewer’s suggestion, we have made some changes in the figures. The colour of the columns has been changed in each graph to help the reader distinguish the treatments applied. In addition, the information in the vertical label has been shortened.

Comment 6: Please correct typos (i.i. apoptosis instead of apopstosis at 4.5).

Reply: Typos and misspellings have been corrected.

Reviewer 2 Report

The original paper of Estaraset al. The authors took up a very interesting topic. I have some suggestions which may increase the quality of this article:

- The introduction should contain the hypothesis of the study.

- Chapter 4.8. Statistical analysis is not properly written. The authors did not inform us about the software they used. There is not any information about the normality of the data. Was is it checked? Do all data have a normal distribution?

- In my opinion, the discussion may benefit from adding some information about the molecular circadian clock and its relationship with hypoxia

- the strengths and limitations of this study should be collected in the "6. Strength and limitations" section.

- Figure 7: BioRender is an easy tool, which allows the preparation of amazing Figures. In my opinion, you should use more options available in this software to make Figure 7 much more "catchy" for reader.

- 496: the link should be cited as a regular citation

Author Response

Manuscript: ID ijms-1203381.

Title: Melatonin Induces Apoptosis and Modulates Cyclins Expression and MAPKs Phosphorylation in Pancreatic Stellate Cells Subjected to Hypoxia.

Authors: Matias Estaras et al.

Reply to the reviewer #2 comments

General comment: The original paper of Estaraset al. The authors took up a very interesting topic. I have some suggestions which may increase the quality of this article.

Reply: Thank you very much for reviewing our manuscript, for your suggestions and for drawing our attention to the different points raised.

Comment 1: The introduction should contain the hypothesis of the study.

Reply: Thank you very much for this comment. We have included the hypothesis of our study in the Introduction section (lines 84-86 and 90-92).

Comment 2: Chapter 4.8. Statistical analysis is not properly written. The authors did not inform us about the software they used. There is not any information about the normality of the data. Was is it checked? Do all data have a normal distribution?

Reply: Information regarding this comment has been included in the Materials and methods section. Normality of data was analysed using Shapiro–Wilk test. The Mann–Whitney U test was used for the statistical analysis and only P values < 0.05 were considered statistically significant. The software employed was GraphPad PRISM® (version 6.01) (lines 615-617).

Comment 3: In my opinion, the discussion may benefit from adding some information about the molecular circadian clock and its relationship with hypoxia.

Reply: We agree with the reviewer´s observation and are grateful for his/her suggestion.We have included in the discussion information about the molecular circadian clock and its relationship with hypoxia (lines 463-479). New bibliography has been cited and included in the bibliography list.

Comment 4: The strengths and limitations of this study should be collected in the "6. Strength and limitations" section.

Reply: Thank you very much for this comment. We have included this new section in the revised version of the manuscript. Lines 654-665.

Comment 5: Figure 7: BioRender is an easy tool, which allows the preparation of amazing Figures. In my opinion, you should use more options available in this software to make Figure 7 much more "catchy" for reader.

Reply: Following the reviewer´s suggestion we have worked to improve figure 7.

Comment 6: 496: the link should be cited as a regular citation.

Reply: Thank you very much for this comment. The link (now line 605) has been placed as regular citation (Ref. 85) in the bibliography list.

Reviewer 3 Report

Ref: ijms- 1203381

Title: Melatonin Induces Apoptosis and Modulates Cyclins Expression and MAPKs Phosphorylation in Pancreatic Stellate Cells Subjected to Hypoxia (Journal: IJMS, MDPI)

Recommendation: Major review

  1. Please provide the information about the ‘n’ number, the number of replicates, number of passages etc. Generally, the Materials and Methods section should be written more detailed.
  2. The Authors should discussed if their study may have translational value to the clinic? The Authors use high concentration of melatonin. Is it potentially safe to the healthy cells?
  3. Almost whole paper is based on wb analyses. Maybe Authors could add some experiments based on apoptosis / necrosis enzymes activities.
  4. Please add the statement that the study was performed in the compliance with EU bioethical law.
  5. Paragraph 4.5 title – misspelling - apoptosis.

Author Response

Manuscript: ID ijms-1203381.

Title: Melatonin Induces Apoptosis and Modulates Cyclins Expression and MAPKs Phosphorylation in Pancreatic Stellate Cells Subjected to Hypoxia.

Authors: Matias Estaras et al.

Reply to the reviewer #3 comments

Thank you very much for reviewing our manuscript, for your suggestions and for drawing our attention to the different points raised.

Comment 1: Please provide the information about the ‘n’ number, the number of replicates, number of passages etc. Generally, the Materials and Methods section should be written more detailed.

Reply: Thank you very much for this observation. The number of passages of the cells was kept to a minimum (at most one passage was performed). This has been mentioned in the Materials and methods section (in 4.2. Culture of Pancreatic Stellate Cells) lines 524-525. With regard to the number of replicates for the different determinations, the data is given in the legends to figures. Additionally, we have written in more detail the Materials and Methods section.

Comment 2: The Authors should discuss if their study may have translational value to the clinic? The Authors use high concentration of melatonin. Is it potentially safe to the healthy cells?

Reply: Thank you very much for this observation. We have included in the discussion information about the actions of pharmacological concentrations of melatonin on healthy cells (lines 453-462).

Comment 3: Almost whole paper is based on wb analyses. Maybe Authors could add some experiments based on apoptosis / necrosis enzymes activities.

Reply: It is right, as the reviewer indicates, that Western blotting analysis represent an important part of the manuscript. However, in the study we also have carried out determinations other than Western blotting. Cell viability assay is based on a colorimetric protocol and employs a microplate reader. The test based on BrdU incorporation into the DNA of growing cells, employed to study cellular proliferation, involves changes in the absorbance of the samples and uses a microplate reader. Additionally, the determination of capase-3 is based on the detection of a reagent that is released upon enzymatic cleavage and is measured by flow cytometry. This reagent is based on a specific binding DNA dye linked to a short peptide (DEVD). In the absence of activation, the reagent is intrinsically non-fluorescent. When apoptosis is activated, caspase-3 cleaves DEVD, enabling the dye to bind to DNA and then produces a bright, fluorogenic response.

Comment 4: Please add the statement that the study was performed in the compliance with EU bioethical law.

Reply: Thank you very much for this comment. This has been mentioned in the Materials and methods section (in 4.2. Culture of Pancreatic Stellate Cells). It states “The mentioned guidelines comply with EU bioethical law”. Lines 529-530.

Comment 5: Paragraph 4.5 title – misspelling - apoptosis.

Reply: Misspelling has been corrected.

Round 2

Reviewer 2 Report

Authors address all the comments.

Author Response

Thank you very much for reviewing our manuscript, for your suggestions, which helped us to improve the quality of our work.

Reviewer 3 Report

The Authors answered all my concernes. Therefore, I recommend this paper to be published as it is.

Author Response

(The authors gave the same response as above.)
